# H3K9 Promotes Under-Replication of Pericentromeric Heterochromatin in Drosophila Salivary Gland Polytene Chromosomes

**DOI:** 10.3390/genes10020093

**Published:** 2019-01-29

**Authors:** Robin L. Armstrong, Taylor J. R. Penke, Samuel K. Chao, Gabrielle M. Gentile, Brian D. Strahl, A. Gregory Matera, Daniel J. McKay, Robert J. Duronio

**Affiliations:** 1Curriculum in Genetics and Molecular Biology, University of North Carolina, Chapel Hill, NC 27599, USA; rarmst02@email.unc.edu (R.L.A.); taylorpenke@gmail.com (T.J.R.P.); gmarieg@live.unc.edu (G.M.G.); brian_strahl@med.unc.edu (B.D.S.); matera@unc.edu (A.G.M.); dmckay1@email.unc.edu (D.J.M.); 2Department of Biology, University of North Carolina, Chapel Hill, NC 27599, USA; samuelkevinchao@gmail.com; 3Department of Biochemistry and Biophysics, University of North Carolina, Chapel Hill, NC 27599, USA; 4Lineberger Comprehensive Cancer Center, University of North Carolina, Chapel Hill, NC 27599, USA; 5Department of Genetics, University of North Carolina, Chapel Hill, NC 27599, USA; 6Integrative Program for Biological and Genome Sciences, University of North Carolina, Chapel Hill, NC 27599, USA

**Keywords:** Endoreplication, under-replication, Drosophila, H3K9, heterochromatin, H4K16

## Abstract

Chromatin structure and its organization contributes to the proper regulation and timing of DNA replication. Yet, the precise mechanism by which chromatin contributes to DNA replication remains incompletely understood. This is particularly true for cell types that rely on polyploidization as a developmental strategy for growth and high biosynthetic capacity. During *Drosophila* larval development, cells of the salivary gland undergo endoreplication, repetitive rounds of DNA synthesis without intervening cell division, resulting in ploidy values of ~1350C. S phase of these endocycles displays a reproducible pattern of early and late replicating regions of the genome resulting from the activity of the same replication initiation factors that are used in diploid cells. However, unlike diploid cells, the latest replicating regions of polyploid salivary gland genomes, composed primarily of pericentric heterochromatic enriched in H3K9 methylation, are not replicated each endocycle, resulting in under-replicated domains with reduced ploidy. Here, we employ a histone gene replacement strategy in *Drosophila* to demonstrate that mutation of a histone residue important for heterochromatin organization and function (H3K9) but not mutation of a histone residue important for euchromatin function (H4K16), disrupts proper endoreplication in *Drosophila* salivary gland polyploid genomes thereby leading to DNA copy gain in pericentric heterochromatin. These findings reveal that H3K9 is necessary for normal levels of under-replication of pericentric heterochromatin and suggest that under-replication at pericentric heterochromatin is mediated through H3K9 methylation.

## 1. Introduction

Proper genome duplication is essential for normal development and tissue homeostasis. In diploid cells, genome duplication and cell proliferation occur via canonical G1→S→G2→M cell cycles in which origins of replication are specified during G1 phase, DNA replication occurs during S phase and chromosome segregation and cell division occur during M phase [1]. Many diploid organisms, including humans, contain tissues composed of cell types with a polyploid genome, a phenomenon called endopolyploidy that serves as a developmental strategy for tissue growth and generating cells with high biosynthetic capacity [2,3,4]. Although endopolyploidy is a common feature of normal development in both plants and animals, polyploidy can also result from mis-regulation of the canonical diploid cell cycle and is commonly associated with human disease [4,5]. Therefore, determining mechanisms that regulate polyploid cell cycles is important for understanding both normal and pathological development.

Polyploidy often arises from endoreplication, a cell cycle in which repetitive rounds of DNA replication occur without intervening mitosis and cell division [2,3,4,5,6,7,8]. The giant polytene chromosomes of the polyploid cells of the *Drosophila* larval salivary gland have long served as a model experimental tissue for understanding endoreplication [8,9]. During late embryonic and larval development, approximately ten endoreplication cycles yield a final ploidy of ~1350C in salivary gland cells [9,10]. These polyploid cells use the same *trans*-acting factors as diploid cells to control DNA replication initiation [9], which occurs stochastically from many origins of replication throughout S phase, yielding reproducibly earlier and later replicating domains [11]. However, replication is not uniform across the salivary gland polyploid genome as it is in diploid cells. Whereas the earlier replicating regions of the genome are duplicated each endocycle, the latest replicating regions are not replicated each endocycle, resulting in under-replicated domains [10,12]. Stalled replication forks at these under-replicated domains cause DNA damage, resulting in deletions that contribute to copy number reduction (<1350C) [13,14]. Under-replicated domains also occur in mammalian polyploid cells [15] and share characteristics of mammalian diploid cell fragile sites [7]. Thus, the study of the origin and properties of under-replicated domains in polyploid genomes will help us understand general features of genome organization and stability.

The mechanistic basis for under-replication is not completely understood but recent studies suggest contributions from regulating origin firing and replication fork progression [7,16]. In both diploid and polyploid cells the timing of DNA replication initiation during S phase correlates with chromatin organization: transcriptionally active, accessible euchromatin generally replicates earlier during S phase whereas transcriptionally repressive, inaccessible heterochromatin generally replicates later [17,18,19,20,21,22,23]. This differential replication timing is regulated, in part, by non-uniform distribution of origins of replication throughout the genome. Euchromatic regions of the genome have a higher density of origins relative to heterochromatic regions, resulting in a higher probability of DNA replication initiation in euchromatin relative to heterochromatin [16,24,25,26]. However, a paucity of origins is an insufficient explanation for under-replication in polyploid cells because some regions of the genome that constitutively lack origins are not under-replicated in all *Drosophila* polyploid cell types [16,27]. Rather, reduced origin density coupled with inhibition of replication fork progression contribute to under-replication within polyploid genomes [16,27,28,29]. The latest replicating regions located within pericentric heterochromatin experience the greatest degree of under-replication. 

Several heterochromatin-associated proteins contribute to under-replication in the *Drosophila* salivary gland. Heterochromatin Protein 1a (HP1a) binds di- and tri-methylated H3K9, which is enriched in pericentric heterochromatin and facilitates heterochromatin formation through multimerization of HP1a molecules and recruitment of other heterochromatin-associated factors [30,31,32]. The SuUR (Suppressor of Under-Replication) protein, a SNF2-like component of silent chromatin in both diploid and polyploid cells [33], localizes to late replicating heterochromatin and inhibits DNA replication fork progression to promote under-replication [28,33,34,35]. HP1a and SuUR recruitment to *Drosophila* salivary gland chromosomes are interdependent on one another: both the absence and over-expression of HP1a disrupt SuUR chromatin binding and over-expression of SuUR results in mis-localization of HP1a to ectopic SuUR sites [36]. Furthermore, tethering either SuUR or HP1a to earlier replicating regions of salivary gland polytene chromosomes is sufficient to delay replication but not to induce under-replication [20]. Rif1 (Rap1-Interacting Factor 1) and the linker histone H1 both directly interact with SuUR and are required for under-replication [29,37]. H1 functions upstream of SuUR and is required for SuUR chromatin binding [37]. Furthermore, although Rif1 directly regulates replication fork progression in a SuUR-dependent manner, SuUR localization to under-replicated regions is independent of Rif1 [29].

In contrast to our current understanding of the contributions of *trans*-acting factors, the roles in endoreplication and under-replication of individual histone tail residues that impact chromatin organization have not been determined. The ninth lysine on histone H3 (H3K9) and the sixteenth lysine on histone H4 (H4K16) have been implicated in promoting S phase progression through studies that modulate factors that catalyze (writers) or bind (readers) the post-translational modifications of these residues [38,39,40,41]. Notwithstanding their importance, these studies cannot directly determine whether H3K9 and H4K16 themselves regulate S phase progression, as histone modifying enzymes also modify non-histone substrates [42,43,44]. To address this issue, we employed a strategy in *Drosophila* to generate histone mutant genotypes, an approach that is not currently feasible in other animals due to the large number of replication-dependent histone genes. The strategy involves deleting the endogenous wild type histone genes and replacing them with transgenic copies encoding an amino acid substitution that prevents post-translational modification of a particular histone residue [45,46].

We recently demonstrated that, in contrast to mutation of H3K9 writers, readers and erasers, *H3K9R* mutant *Drosophila* diploid wing imaginal discs have only a modestly reduced S phase index, suggesting a small role played by H3K9 in canonical S phase progression [21]. We observed a similarly modest effect on S phase progression in *H4K16R* wing imaginal disc cells [47]. Here, we utilize replication-dependent *H3K9R* and *H4K16R* mutations to probe the role of heterochromatin and euchromatin, respectively, in cell cycle phasing, DNA replication timing and under-replication in *Drosophila* salivary gland polytene chromosomes. We demonstrate that H3K9 regulates endoreplication whereas H4K16 is largely dispensable. Furthermore, we demonstrate that H3K9 promotes under-replication of pericentric heterochromatin whereas under-replication along chromosome arms is H3K9-independent. 

## 2. Materials and Methods

### 2.1. Drosophila Larval Culturing 

All fly stocks were maintained on standard corn medium and crossing schemes to generate engineered replication-dependent histone genotypes were performed as in Reference [21]. Fifty GFP-positive Histone Wild Type (*HWT*), *H3K9R* or *H4K16R* larvae were cultured independently of their phenotypically wild type, GFP-negative siblings by manually moving first-instar larvae into vials of standard corn medium and allowing them to develop until third instar larvae. Note that only the replication-dependent *H3* and *H4* histone genes were mutant in this study.

### 2.2. Salivary Gland Polytene Chromosome Immunofluorescence

Pre-wandering third-instar larvae were staged using the following criteria: crawling on top of the media, not displaying wandering behavior on vial edges and no longer eating. Salivary glands were dissected from pre-wandering third-instar larvae in 1× PBT (0.1% Triton X-100 in PBS, pH 7.5). Glands were permeabilized and fixed in the following solutions: (1) 2 min in (3.7% Paraformaldehyde, 1× PBT), (2) 2 min in (3.7% Paraformaldehyde, 50% Acetic Acid) and moved to (3) 1:2:3 Lactic Acid: dH_2_O: Acetic Acid on a siliconized coverslip. Spread polytene chromosomes were flash frozen in liquid N_2_ and stored in 1× PBT until all slides were completed. Polytene chromosome spreads were incubated in Image-iT FX Signal Enhancer (Thermo Fisher) for 30 min at room temperature, stained with 1:500 mouse anti-PCNA (Abcam; ab29) and 1:1500 mouse anti-HP1a (DSHB; C1A9) in 1× PBT overnight at 4 degrees, treated with 0.2µg/mL DAPI for 5 min at room temperature, mounted in ProLong Gold antifade reagent and imaged on a Leica confocal microscope. PCNA patterns were staged according to the following criteria: ER) dim PCNA signal across chromosome arms with few gaps between bands of PCNA signal; E-MR) bright PCNA signal across chromosome arms with few, distinct gaps between bands of PCNA signal; M-LR) thick bands of PCNA signal across chromosome arms with large gaps between bands of PCNA signal and bright chromocenter PCNA signal; LR) thin bands of PCNA signal across chromosome arms with large gaps between bands of PCNA signal and chromocenter PCNA signal; VLR) dim, sparse PCNA signal primarily at the chromocenter; NR) no PCNA signal.

### 2.3. Sample Preparation for Genome Sequencing

Salivary glands from female, wandering third-instar larvae were isolated and flash frozen in liquid N_2_ until all samples were collected. Nuclei were isolated from replicates of 25 salivary glands and sonicated with a Branson Sonifier 450 to an average fragment size distribution of 500–1000 bp. Sonicated samples were treated with 100 µg/mL RNaseA at 37 °C for 1 h and with 200 µg/mL proteinase K at 37 °C for 2 h. Genomic DNA was phenol chloroform extracted and stored at −80 °C prior to library preparation. Libraries were prepared from 20 ng of genomic DNA with the ThruPLEX DNA-seq kit (Rubicon Genomics) and sequenced on an Illumina HiSeq2500 at the UNC High-Throughput Sequencing Core Facility. Sequencing data can be obtained using GEO accession number GSE125505.

### 2.4. Bioinformatics

Paired-end 100bp reads were trimmed using Trimmomatic (v0.36) with LEADING:30 and TRAILING:30 parameters and aligned to the dm6 reference genome (release 6.04) using Bowtie2 (v2.3.2) default parameters. Reads with a MAPQ score greater than 10 were retained with SAMtools (v1.6), which removes reads with low confidence mappability that often include simple repeats. Thus, reads within heterochromatin map uniquely; see [21,48,49]. BEDTools coverage (v2.25.0) was used to quantify the number of reads mapping to 10kb windows tiled across the genome, with results normalized to read depth [50]. CNVnator 0.3.3 was used for the detection of under-replicated sequences using a bin size of 1000 [51]. Under-replicated domains were called if they were: (1) detected by CNVnator and (2) greater than 10kb in size. Pericentromeric and chromosome arm boundaries were defined by high levels of H3K9me2 enrichment as in References [52,53]. A p value of 0.001 was used as the statistical significance cutoff for all bioinformatic analyses. modENCODE H3K9me2 ChIP-seq data from whole third instar larvae (GEO accession number GSE47260) was used for analyses and LOESS regression lines were generated with span = 0.02.

## 3. Results

We used our histone gene replacement platform to generate *H3K9R* and *H4K16R* mutant larvae and determine whether replication-dependent H3K9 and H4K16 are necessary for endoreplication in the *Drosophila* salivary gland. The approximately ten salivary gland endoreplication cycles occur from 7–96 h after egg deposition and are completed by the wandering third-instar larval stage of development [9,10]. To visualize nuclei actively undergoing DNA replication, we prepared salivary gland polytene chromosome spreads from pre-wandering third-instar larvae and stained them with an antibody against Proliferating Cell Nuclear Antigen (PCNA), as done previously [11,37]. PCNA travels with the replisome and functions as a DNA polymerase processivity factor [54] and thus serves as a cytological marker of active replication forks [55]. Consequently, PCNA-positive polytene chromosome spreads are undergoing endo-S phase whereas PCNA-negative polytene chromosome spreads are not (i.e., G phase) (Figure 1A).

As a general assessment of endocycle progression, we first determined an S phase index for each mutant genotype. Whereas approximately 60% of Histone Wild Type control (*HWT*; *n* = 211) and *H4K16R* mutant (*n* = 284) polytene chromosome spreads were in S phase (i.e., PCNA-positive) (Figure 1B), only ~30% of *H3K9R* (*n* = 286) mutant salivary glands were in S phase (Figure 1B; *p* < 0.0001). These data suggest that H3K9 is required for proper endocycle progression in the *Drosophila* salivary gland whereas H4K16 is dispensable. Consistent with this interpretation, *H3K9R* mutant salivary glands are smaller than *HWT* control glands at early larval stages (Figure 1C). However, because development is delayed by ~24 h in *H3K9R* mutants [48], they eventually attain a similar size as *HWT* control glands by the pre-wandering stage (Figure 1C).

In the salivary gland polytene chromosome spreads, the largely euchromatic chromosome arms extend from a single chromocenter composed of the pericentric heterochromatin from each of the four chromosomes (Figure 1A). Genome-wide patterns of active replication change throughout the duration of S phase, with chromosome arms replicating earlier and the chromocenter replicating later. In addition, the euchromatin along chromosome arms (DAPI-dim inter-bands; Figure 1E) replicates earlier than intercalary heterochromatin (DAPI bright bands; Figure 1E), resulting in changing patterns of PCNA staining that can be used to monitor S phase progression. As performed previously [28,37], we binned polytene chromosome spreads from pre-wandering third-instar larvae into one of six RT categories based on the pattern of active replication as determined by PCNA staining: early-replicating (ER), early/mid-replicating (E-MR), mid/late-replicating (M-LR), late-replicating (LR), very late-replicating (VLR) and non-replicating (*n*R) (Figure 1A; Materials and Methods). If either histone mutation affected endoreplication, such as the time during S phase when either euchromatin or heterochromatin replicates (i.e., replication timing or RT), we would expect a change in PCNA staining patterns and/or a change in the distribution of these categories relative to control.

H4K16 acetylation is found in euchromatic regions of the genome while H3K9 methylation is enriched in heterochromatin [56]. Consequently, *H4K16R* and *H3K9R* mutants might influence earlier and later replicating regions of the genome, respectively. However, we recently demonstrated that genome-wide RT in female diploid wing imaginal disc cells is unchanged in *H4K16R* mutants [21]. Furthermore, pericentric heterochromatin in these cells generally remains late replicating in *H3K9R* mutants, although select domains within the pericentromeres advance replication timing [21]. Despite modest RT effect in diploid cells of these histone mutant wing discs, we next asked whether H3K9 and H4K16 directly contribute to RT during endoreplication in the salivary gland. Our single-blind analysis revealed a significant difference in RT pattern distributions between both *H3K9R* (*n* = 65; *p* < 0.00001) and *H4K16R* (*n* = 101; *p* = 0.004956) when compared to *HWT* control (*n* = 96) (Figure 1D). The *H4K16R* and *HWT* RT distributions were similar, with M-LR as the most prevalent RT category in each genotype. However, the *H4K16R* RT distribution shifts from M-LR to earlier replicating patterns (ER and E-MR) relative to *HWT* control (Figure 1D). In contrast, the M-LR RT category was the least prevalent in *H3K9R* mutants and the overall distribution of *H3K9R* RT categories was obviously different than either *HWT* or *H4K16R*. This distribution shows a biphasic shift, with increases in both earlier (ER and E-MR) and VLR RT patterns relative to *HWT* control (Figure 1D). High magnification images revealed that for both histone mutants PCNA staining in early-mid S phase chromosomes occurs in DAPI-dim inter-bands and in late S phase PCNA staining occurs in DAPI-bright bands, as in the *HWT* control (Figure 1E). These data suggest that the timing of replication of intercalary heterochromatin is not advanced in either histone mutant and that large-scale changes in RT are not solely responsible for the change in distribution of RT categories.

We posited that the increased proportion of *H3K9R* mutant chromosomes with a VLR PCNA pattern (where replication is occurring primarily at the chromocenter) represents more extensive replication of normally under-replicated sequences at the pericentromeres. This hypothesis predicts that following completion of all endoreplication cycles, *H3K9R* mutants would have increased copy number at normally under-replicated sequences compared to both *H4K16R* mutants and *HWT* controls. To test this hypothesis, we subjected genomic DNA isolated from *H3K9R*, *H4K16R* and *HWT* wandering third-instar larval salivary glands to Illumina sequencing (Figure 2A–C; Appendix A). At this developmental stage, endoreplication cycles have ceased and salivary gland cells have reached their final ploidy. DNA copy number profiles from two biological replicate samples were generated by determining the normalized read count of paired-end 100bp reads at 10kb windows tiled across the genome. The replicate samples from each of the three genotypes correlated well with each other (Appendix A). Therefore, for all subsequent analyses, we used the averaged reads per million (RPM) normalized values of the two replicates per genotype (Appendix A).

To identify DNA copy number differences genome-wide, we determined normalized copy number values at 10 kb windows along all major chromosome scaffolds (2L, 2R, 3L, 3R, 4 and X) for *H3K9R* or *H4K16R* and *HWT* control (Appendix A). The data were plotted as a log_2_ transformed mutant/*HWT* ratio of normalized read counts, resulting in a relative copy number change for each 10kb window (Figure 2C). The most striking feature of these data is an increase in DNA copy number at *H3K9R* pericentromeres (*p* < 2.2 × 10^−16^; Student’s *T* test) (Figure 2A,C; Appendix A). Some *H3K9R* pericentric regions are enriched sixteen-fold relative to *HWT* control. These whole genome sequencing data suggest that increased copy number at pericentromeres may contribute to the observed disorganization and enlargement of *H3K9R* chromocenters that we reported previously (Figure 2D) [48]. In addition, when considered as a whole, the DNA copy number along chromosome arms in *H3K9R* mutants shows a slight but statistically significant (*p* < 2.2 × 10^−16^), decrease relative to *HWT* control (Figure 2C). This decrease may result from a larger proportion of *H3K9R* mutant reads mapping to pericentric regions, causing a corresponding relative decrease in the number of reads for other genomic regions and thus is not likely biologically meaningful. In contrast to the *H3K9R* results, we find that DNA copy number in *H4K16R* mutants is not significantly different than *HWT* controls either at pericentric regions or along chromosome arms (Figure 2B,C; Appendix A). In agreement with these data, the *H4K16R* chromocenter is cytologically similar to the *HWT* control (Figure 2D,E). 

These whole genome sequencing data suggest that *H3K9R* mutants, but not *H4K16R* mutants, are defective in under-replication. To assess this possibility more directly, we next identified under-replicated domains in *H3K9R*, *H4K16R* and *HWT* using CNVnator, a highly sensitive method for copy number variation detection based on a statistical analysis of read depth of short reads [51]. This method was used previously in *Drosophila* [29]. We required that under-replicated domains called by CNVnator be greater than 10 kb. Using this criterion, we detected 101 under-replicated domains in *HWT* control salivary glands, 86 and 98 of which overlap with domains identified in *H3K9R* and *H4K16R* mutants, respectively. The size distribution of under-replicated domains in each of the three genotypes were similar, with sizes ranging from 11 kb to over a megabase and medians ranging from 127 kb to 138 kb (Figure 3A). 

We used the 101 under-replicated domains identified in *HWT* salivary glands to more closely investigate under-replication in *H3K9R* and *H4K16R* mutants. We also included in our analysis previously published Illumina sequencing data from *SuUR* mutant salivary glands, which have reduced under-replication both at pericentromeres and along chromosome arms [35]. When we use 10kb windows to compare DNA copy number at under-replicated domains between each mutant genotype and their respective controls, we observe a significant reduction of under-replication in both *SuUR* and *H3K9R* mutants (*p* < 2.2 × 10^−16^; Student’s *T* test) (Figure 3B). In addition, we found a small but statistically significant decrease in copy number at fully replicated regions in *H3K9R* mutants (*p* = 8.25 × 10^−7^; Student’s *T* test), which as noted above most likely results from the way the sequencing data was analyzed rather than a biological phenomenon (Figure 3B). In contrast, we found no significant changes in copy number at either subset of genomic regions in *H4K16R* mutants (Figure 3B). These data suggest that copy number differences between *H3K9R* and *HWT* are due to failure of the normal under-replication mechanism. 

We next partitioned the 101 under-replicated regions identified in *HWT* into those located on chromosome arms and those located within pericentric heterochromatin. For this purpose, pericentric heterochromatin was defined by high levels of H3K9me2 enrichment as described previously [52,53]. When considering all under-replicated domains within pericentric heterochromatin, we observed a significant increase in copy number in *H3K9R* mutants (*p* < 2.2 × 10^−16^) relative to *HWT* control (Figure 3C,D). Furthermore, 10/48 (21%) under-replicated domains identified in *HWT* pericentric heterochromatin were not identified using CNVnator in *H3K9R* mutants, suggesting a strong suppression of under-replication in these 10 domains. When considering copy number at under-replicated domains along chromosome arms, we observed a small but statistically significant (*p* < 4.429 × 10^−10^) decrease in copy number in *H3K9R* mutants relative to *HWT* control (Figure 3C,D). This decrease in copy number may result from the 18 newly identified under-replicated domains in *H3K9R* that were not identified in *HWT* or may result from the sequencing analysis as noted above. In contrast to these data, *SuUR* mutants have reduced under-replication both at pericentric heterochromatin and along chromosome arms (Figure 3C) [29,35]. We found no significant changes in copy number either at under-replicated regions in pericentric heterochromatin or along chromosome arms in *H4K16R* mutants (Figure 3C,D). These data demonstrate that H3K9 is required for normal under-replication at pericentric heterochromatin and suggest that under-replication along chromosome arms occurs in an H3K9-independent manner.

## 4. Discussion

Here, we utilized a genetic platform for histone gene replacement to interrogate the function of replication-dependent H3K9 and H4K16 in *Drosophila* salivary gland endoreplication. We found that while H3K9 is important for salivary gland endoreplication, H4K16 is largely dispensable. We observed three phenotypes in *H3K9R* mutant salivary glands: (i) a decrease in the S phase index, (ii) a biphasic shift in replication timing toward both earlier and the very latest (i.e., chromocenter replication) patterns and (iii) a reduction in the level of under-replication at pericentric heterochromatin but not along chromosome arms. 

A decrease in the S phase index could result from a reduction in the number or duration of endo-S phases, an increase in the duration of G phase, or both. The *H3K9R* mutant salivary glands reach the same size as control, consistent with completion of the endoreplication program. In addition, the slower development of *H3K9R* mutant animals may result in longer G periods between endo S phases, thus further reducing the S phase index. Another possibility is that the endoreplication timing program is generally condensed in *H3K9R* mutants such that S phase occurs more quickly, accounting for the change in distribution of RT categories and contributing to the decrease in S phase index. These changes unlikely indirectly result from changes in transcription, as we did not detect significant changes in the expression of protein coding genes, including those encoding replication factors, in the replication-dependent *H3K9R* mutant [21,48]. 

The latest replicating sequences of polyploid salivary gland cells are not fully replicated each endocycle, yielding regions of the genome with decreased copy number, particularly in pericentric heterochromatin but also at specific loci along chromosome arms [12]. We suggest that the elevated number of *H3K9R* mutant salivary gland nuclei with a very late replication pattern, represented by PCNA staining of the chromocenter, results from more extensive replication of pericentric heterochromatin each endocycle. Consistent with such a failure of normal under-replication, in *H3K9R* mutants we detected an increase in DNA copy number at pericentric heterochromatin by whole genome sequencing, as well as altered chromocenter cytology [48]. Interestingly, we did not detect decreased under-replication along chromosome arms, indicating that replication-dependent H3K9 is particularly important for endoreplication control in pericentric heterochromatin. This result is reminiscent of our previous observations that replication-dependent *H3K9R* mutation disrupts HP1a recruitment, nucleosome occupancy and transposon repression at pericentric heterochromatin in diploid wing imaginal discs, without appreciably affecting the function of euchromatin [21,48,49]. An important caveat to our observations is that we cannot rule out a contribution from K9 of the variant histone H3.3 to under-replication along salivary gland polytene chromosome arms. Animals in which both H3.3 and replication-dependent H3 contain K9R mutations die as early first instar larvae, precluding the appropriate genetic experiment [49]. Similarly, our *H4K16R* analyses cannot rule out a small contribution from the replication-independent *His4r* gene, which resides outside the replication-dependent histone gene cluster and encodes a protein identical to replication-dependent H4 [21,57].

Our previous analysis of *H3K9R* diploid wing discs revealed only a small number of advanced RT changes within pericentric heterochromatin [21]. In contrast, our analyses here revealed that most under-replicated domains within pericentric heterochromatin increased in DNA copy number in *H3K9R* mutant polyploid salivary glands. These data suggest that replication-dependent H3K9 plays a more significant role in regulating under-replication during salivary gland endoreplication than in regulating late replication during the canonical diploid cell cycle. Alternatively, these processes might be controlled by distinct mechanisms. The biological function of under-replication is not known and thus the consequence of losing pericentric under-replication is uncertain. Interestingly, *Rif1* mutants, which lack under-replication altogether [29], are viable and fertile [29,58].

What might be the mechanism by which H3K9 promotes under-replication of pericentric heterochromatin? Previous studies established SuUR as a key regulator of under-replication in polyploid genomes of *Drosophila* salivary glands [20,29,33,34,35]. Unlike *H3K9R* mutants, under-replication in *SuUR* mutants is reduced both along chromosome arms and at pericentric heterochromatin [33]. The mode of SuUR association with these two regions of the genome is different, being SNF2 domain-dependent and dynamic with replication forks and SNF2 domain-independent and more constitutive within pericentric heterochromatin [29]. SuUR forms a protein complex with Rif1, which recruits Protein Phosphatase 1 (PP1) and these interactions are required to promote under-replication [29]. In addition, HP1a and SuUR depend on one another for chromatin association [20,36]. Thus, one potential explanation for reduced pericentric heterochromatin under-replication in *H3K9R* mutants is the loss of pericentric HP1a [48]. In the absence of HP1a, SuUR’s constitutive association with pericentric heterochromatin is reduced [36], which may prevent the downstream effectors of under-replication, Rif1 and PP1, from properly suppressing replication at these regions of the genome.

In conclusion, our data indicate that under-replication at salivary gland pericentric heterochromatin occurs through an H3K9-dependent mechanism and therefore suggest that the hallmarks of constitutive heterochromatin, H3K9me and HP1a, are critical regulators of under-replication in *Drosophila* polyploid cells. 

## Figures and Tables

**Figure 1 genes-10-00093-f001:**
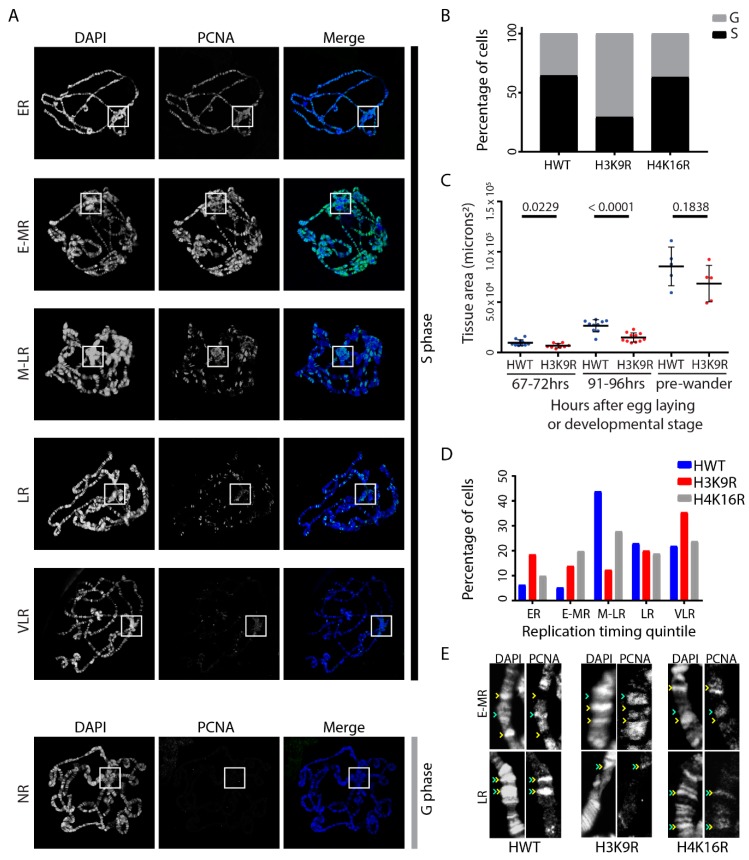
H3K9 promotes endoreplication of the *Drosophila* salivary gland. (**A**) Polytene chromosome spreads from *HWT* pre-wandering third-instar larvae stained for PCNA (green) and DAPI to detect DNA (blue). Representative early-replicating (ER), early/mid-replicating (E-MR), mid/late-replicating (M-LR), late-replicating (LR), very late-replicating (VLR) and non-replicating (*n*R) PCNA patterns are shown. White boxes designate the chromocenter as identified by HP1a staining (*HWT* and *H4K16R*) or cytologically (*H3K9R*) (*n*ot shown). (**B**) Percentage of PCNA-positive (S phase; black) and PCNA-negative (G phase; grey) polytene chromosome spreads for *HWT* (*n* = 211), *H3K9R* (*n* = 286; *p* < 0.001) and *H4K16R* (*n* = 284; *p* > 0.05) genotypes. Significance was determined using the Chi-squared test. All S phase measurements were taken at the pre-wandering developmental stage. (**C**) Salivary gland area of *HWT* and *H3K9R* at 67–72 h after egg deposition (*p* = 0.0229), 91–96 h after egg deposition (*p* < 0.0001) and at the pre-wandering third-instar larval stage (*p* = 0.1838) (Student’s *T* test). (**D**) Percentage of PCNA-positive polytene chromosome spreads in each of the five replication timing pattern categories shown in A for *HWT* (*n* = 96), *H3K9R* (*n* = 65; *p* < 0.00001) and *H4K16R* (*n* = 101; *p* = 0.004956) genotypes. Significance determined using the Chi-squared test and *HWT* data as the expected categories. (**E**) Representative chromosome arms with E-MR and LR PCNA-patterns for *HWT, H3K9R* and *H4K16R*. In E-MR patterns, PCNA colocalizes with DAPI-dim inter-bands (green arrowheads) but not DAPI-bright bands (yellow arrowheads). In LR patterns, PCNA colocalizes with DAPI-bright bands (green/yellow double arrowheads).

**Figure 2 genes-10-00093-f002:**
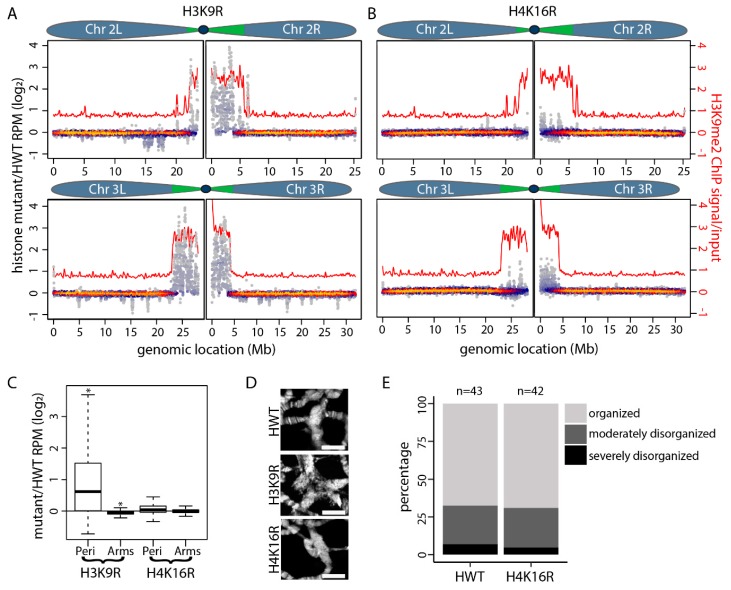
DNA copy number in pericentric heterochromatin is elevated in *H3K9R* mutants. (**A**,**B**) Heatscatter plot of (**A**) *H3K9R*/*HWT* log_2_ ratio and (**B**) *H4K16R*/*HWT* log_2_ ratio of normalized copy number at 10kb windows along Chromosomes 2 and 3. LOESS regression line of modENCODE H3K9me2 ChIP signal is shown in red (GSE47260). (**C**) Quantification of mutant/*HWT* ratio of normalized copy number at 10kb windows for all major chromosome scaffolds (Chromosomes 2L, 2R, 3L, 3R, 4 and X) separated into pericentromeres (Peri) and chromosome arms (Arms) (* = *p* < 0.001; Student’s *T* test). Coordinates for pericentromeres and chromosome arms were defined in References [52,53] (see also Appendix A). (**D**) Representative polytene chromosome chromocenter from *HWT*, *H3K9R* and *H4K16R* wandering third-instar salivary glands stained with DAPI. Scale bar = 10 µM. (**E**) Quantification of cytological categories for *HWT* and *H4K16R* chromocenters as performed in Reference [48]. *HWT* and *H4K16R* chromocenters shown in panel D represent the organized category whereas the *H3K9R* chromocenter shown represents the severely disorganized category, which we previously reported comprises 72% of *H3K9R* chromocenters [48]. The distribution of chromocenters among the three categories between *HWT* and *H4K16R* is not statistically different (*p* > 0.0001; Chi squared test).

**Figure 3 genes-10-00093-f003:**
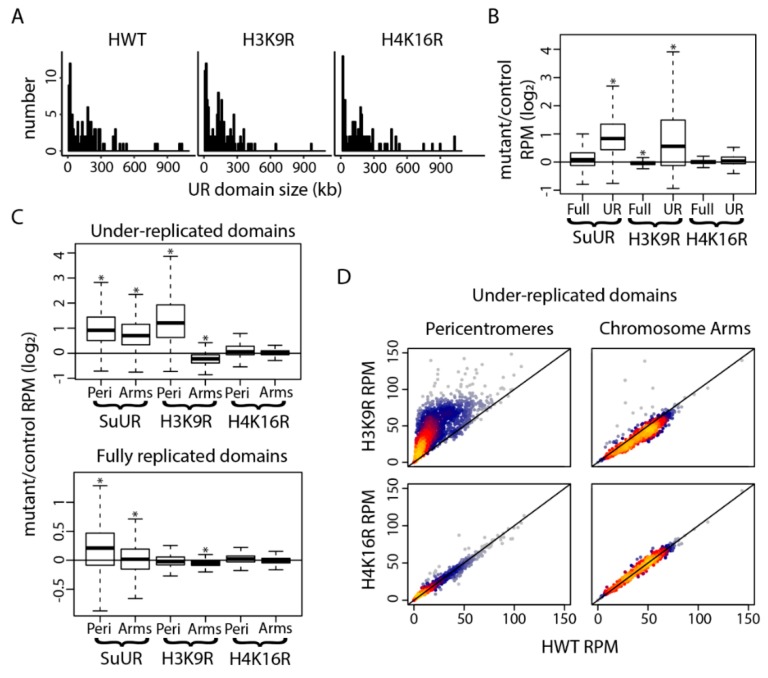
Under-replication of pericentric heterochromatin is H3K9-dependent. (**A**) Histogram of under-replicated domains identified by CNVnator [51] for *HWT*, *H3K9R* and *H4K16R* genotypes. Bin size is set to 10 kb. (**B**) Boxplot of the *SuUR/OregonR*, *H3K9R*/*HWT* and *H4K16R*/*HWT* log_2_ ratios of DNA copy number at 10 kb windows at fully replicated (Full) and under-replicated (UR) domains as defined by CNVnator [51] in *HWT* (* = *p* < 0.001; Student’s *T* test). (**C**) Boxplot of the *SuUR/OregonR*, *H3K9R*/*HWT* and *H4K16R*/*HWT* log_2_ ratios of normalized signal at 10kb windows at under-replicated domains (top panel) and fully replicated domains (bottom panel) in pericentromeres (Peri) or chromosome arms (Arms) (* = *p* < 0.001; Student’s *T* test). (**D**) Heatscatter plot of normalized signal at 10 kb windows at under-replicated domains in *HWT* versus *H3K9R* (top panels) or versus *H4K16R* (bottom panels) at pericentromeres (left panels) and chromosome arms (right panels).

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
