# Peer review of "H3K9 Promotes Under-Replication of Pericentromeric Heterochromatin in Drosophila Salivary Gland Polytene Chromosomes"

_genes, 2019, doi:10.3390/genes10020093_

Round 1
Reviewer 1 Report
In this manuscript, Armstrong et al. use the histone gene replacement system previously established in the Duronio lab to interrogate the roles H3K9 and H4K16 have on endo cycle dynamics, replication timing and copy number number control using the Drosophila larval gland as a model system. They find that H3K9 has a significant affect on these processes, while H4K16 has more modest effects, if any. The most significant finding is that the H3K9mutant results in the loss of underreplication of the pericentric heterochromatin, but not the euchromatic underreplicated regions in the chromosome arms. The paper is concise and very nicely written. The data is of high quality and warrants publication largely as is. I only have minor suggestions listed below.
Pg. 2, Line 56:
Should have better reference here, not just generic RT review. The authors should reference RT in polytene cells specifically.
Pg. 5, Figure 1D:
Are there supposed to be error bars here? I know the stats are mentioned in the text, but it would be helpful to know if these differences are meaningful when looking at the graph.
Pg. 10, Line 368-370
It would be useful here to mention how SUUR has two modes of chromatin binding during the endo cycle. Constitutive to the heterochromatin, while dynamically associating with forks in the arm regions. Seems to parallel the authors observations. Kolesnikova et al., Chromosoma (2013) 122:55–66
Also, SUUR binding to pericentric heterchromatin is reduced in HP1 mutant SG chromosomes. Therefore, this sentence is more than speculation.
see Pindyurin et al., J. Cell Science 2008
Author Response
Pg. 2, Line 56:
Should have better reference here, not just generic RT review. The authors should reference RT in polytene cells specifically.
We agree with the reviewer’s suggestion. The reference has been changed to:
Kolesnikova, T. D., Goncharov, F. P., & Zhimulev, I. F. (2018). Similarity in replication timing between polytene and diploid cells is associated with the organization of the Drosophila genome. PLoS One, 13(4), e0195207. doi:10.1371/journal.pone.0195207
Pg. 5, Figure 1D:
Are there supposed to be error bars here? I know the stats are mentioned in the text, but it would be helpful to know if these differences are meaningful when looking at the graph.
Each bar in the histogram indicates the total number of nuclei in each category, rather than counts from multiple trials. Thus, we used a chi-squared analysis to ask whether the values in each category in the two histone mutant genotypes were significantly different from the expected values, represented by the HWT counts.
Pg. 10, Line 368-370
It would be useful here to mention how SUUR has two modes of chromatin binding during the endo cycle. Constitutive to the heterochromatin, while dynamically associating with forks in the arm regions. Seems to parallel the authors observations. Kolesnikova et al., Chromosoma (2013) 122:55–66
We like this suggestion and have added a sentence the Discussion when we describe mechanisms that might explain our results (lines 385-387, 390).
Also, SUUR binding to pericentric heterchromatin is reduced in HP1 mutant SG chromosomes. Therefore, this sentence is more than speculation.
see Pindyurin et al., J. Cell Science 2008
We thank the reviewer for pointing out how using this citation strengthens our model. The citation is now included in line 391.

Reviewer 2 Report
Armstrong et al. use a genetic system in Drosophila to analyze the effects of H3K9 and H4K16 histone mutants on DNA replication in polyploid cells of the Drosophila salivary glands. In these cells, heterochromatin is enriched for H3K9me3 and reduced in DNA copy number because it does not duplicate every endocycle S phase, so called underreplication. Their major conclusion is that while H3K9 is important for salivary gland endoreplication, H4K16 is largely dispensable. They show that H3K9 mutant cells have a decreased S phase fraction, a shift in the fraction of cells with specific replication timing patterns, and an increase in DNA copy number within pericentric heterochromatic domains but not the intercalary heterochromatic domains on chromosome arms.
This manuscript reports some interesting findings that should be of interest to those in the chromatin and DNA replication fields. The observation that H3K9R mutants affect copy number of pericentric but not intercalary heterochromatic DNA is especially intriguing. A strength is their genetic system that permits analysis of histone mutant residues rather than the enzymes that modify them, which have multiple substrates and pleiotropic effects. I have a number of concerns, however, about interpretation and presentation, some of which can be fixed be editing, but others require further analysis of data.
1) My major concern is how replication timing for H3K9 mutants is presented and interpreted. The analysis depends on previous published work that linked the time of replication to specific patterns of labeling of polytene bands, interbands, and heterochromatin. Here, timing is inferred by these previously defined patterns of labeling. What is observed is an overall reduced S phase fraction and a change in the fraction of cells with specific chromosomal PCNA labeling patterns. If the H3K9R mutant specifically advances timing of heterochromatin domains, however, one would expect a completely new pattern of labeling in which heterochromatin labels together with earlier replicating bands and interbands. From the narrative and figures, it is hard to know if this was not looked for or just not observed. The authors do not show micrographs of labeling patterns for H3K9 or H3K16 mutants. They show graphs for fraction of cells with different labeling patterns, which suggest that H3K9R increases the fraction of cells with the early and very late replication pattern (the latter inferred by labeling of heterochromatic domains only). This suggests that in H3K9R more cells are replicating their heterochromatic DNA, but are still doing so “late” because bands and interbands are not co-labeled.
A) Help the reader by adding a paragraph to the beginning of the results that explains how different patterns of labeling are interpreted as timing.
B) It seems that part of the “timing” interpretation is being driven by the finding that H3K9R has a smaller S phase fraction, suggesting that the duration of S phase is shorter and hence replication timing is advanced for all / most domains. The discussion acknowledges, however, that a smaller S phase fraction could be the result is that G phase is longer, not S phase shorter. Given that in H3K9R more cells have labeling of pericentric heterochromatin only (the “late” pattern) and have an increased heterochromatic DNA copy number, does that mean that replication level and timing have been uncoupled? How does integration of S phase fraction data influence this interpretation? The narrative needs to be improved to explain this logic.
C) Describe whether the PCNA labeling of domains was altered relative to other domains, e.g. whether some nuclei had labeling of heterochromatic domains together with euchromatic bands that replicate earlier.
D) Please discuss how H3K9R mutation could be increasing fraction of cells with early and early-mid labeling pattern in which non-heterochromatic bands and interbands are labeled. Acknowledge in discussion that changes in S phase fraction, and perhaps other phenotypes, may be an indirect effect of H3K9 mutants on transcription.
E) Show select examples of histone mutant polytene labeling.
F) Show higher mag (e.g. insets) of polytene labeling, hard to see bands / interbands in current figures.
2) Figure 1B: It would have been more complete to analyze S phase fraction by whole mount EdU labeling given that polytene squash immunolabeling can be highly variable.
3) Figure 2A: Make more explicitly clear that copy number in H3K9R is normalized relative to wild type salivary glands. The reader may interpret gray dots as rereplication of pericentric heterochromatic DNA above the copy number of euchromatic DNA, rather than less under-replication.
4) Related to 3, can something be said about the absolute copy number of heterochromatic DNA in H3K9R? Is it close to fully duplicated copy number of euchromatic DNA? Was it normalized to control embryonic diploid reads? Copy number of some regions increased 16-fold relative to wild type, but it is unclear which loci those are and how much they are under-replicated in wild type.
5) Say whether all types of pericentric heterochromatic DNA was considered in the sequence analysis. If simple repeats were included, say how these highly repetitive DNA classes were bioinformatically analyzed.
6) If HP1 requires H3K9me3 for localization, and HP1 and Su(UR) localization are co-dependent, why does Su(UR) affect IH underreplication whereas H3K9me3 does not? Is HP1 found in intercalary heterochromatin?
7) Include error bars and p value asterisks within graphs of figure 1. Which classes were significantly different from each other in 1D?
8) In several places “ectopic replication” is mentioned. I don’t think that’s the right term because what is being described is extra replication or changed timing, whereas ectopic means “abnormal place.”
9) Figure 1 legend says the chromocenter was identified by anti-HP1 labeling, but I am assuming that was not done for the mutants since H3K9R disrupts HP1 localization.
Author Response
A) Help the reader by adding a paragraph to the beginning of the results that explains how different patterns of labeling are interpreted as timing.
A description of the criteria for PCNA pattern categories was added to the Materials and Methods.
B) It seems that part of the “timing” interpretation is being driven by the finding that H3K9R has a smaller S phase fraction, suggesting that the duration of S phase is shorter and hence replication timing is advanced for all / most domains. The discussion acknowledges, however, that a smaller S phase fraction could be the result is that G phase is longer, not S phase shorter. Given that in H3K9R more cells have labeling of pericentric heterochromatin only (the “late” pattern) and have an increased heterochromatic DNA copy number, does that mean that replication level and timing have been uncoupled? How does integration of S phase fraction data influence this interpretation? The narrative needs to be improved to explain this logic.
We appreciate the reviewer’s attention to the various complexities when interpreting changes in the distribution of PCNA staining patterns, as well as a change in the S phase index (below). We concur that these types of data are not always easy to interpret, and thus we have tried to be careful in our conclusions, including by not directly linking the two measurements. As suggested by the reviewer, we have reorganized and expanded our introduction of the PCNA staining data to hopefully assist the reader in thinking about these data. We have also now included higher magnification images of H3K9R mutant, H4K16R mutant and HWT control in Figure 1E highlighting the PCNA staining in bands and inter-bands of the polytene chromosomes in late and early-mid S phase, respectively. We now emphasize that our data do not indicate a wholesale change in RT across the genome despite shifts in the numbers of each category. We have tried to make this point more clearly on page 6. We interpret our cytological and whole genome sequencing data as an indication that pericentric heterochromatin replicates more completely in the H3K9R mutants and is still replicating late in endo S phase. We have not assessed whether it begins earlier and thus changes its replication timing.
C) Describe whether the PCNA labeling of domains was altered relative to other domains, e.g. whether some nuclei had labeling of heterochromatic domains together with euchromatic bands that replicate earlier.
We have now added high magnification images of a portion of histone mutant polytene chromosome arms in Figure 1E. These data show that in HWT control and both histone mutants that PCNA staining in early mid S phase occurs in DAPI-dim inter-bands, and in late S phase PCNA staining occurs in DAPI-bright bands (i.e. intercalary heterochromatin). These data suggest that the timing of replication of intercalary heterochromatin is not advanced.
D) Please discuss how H3K9R mutation could be increasing fraction of cells with early and early-mid labeling pattern in which non-heterochromatic bands and interbands are labeled. Acknowledge in discussion that changes in S phase fraction, and perhaps other phenotypes, may be an indirect effect of H3K9 mutants on transcription.
We did not assess closely enough the timing of specific bands and inter-bands to make conclusions regarding the basis for the increase in the earlier replicating categories. In our previously published study we did not observe significant changes in the expression of protein coding genes in the H3K9R mutant (Penke et al. 2016). Thus, such indirect effects on replication changes are unlikely, and we have explicitly made this point in the Discussion.
E) Show select examples of histone mutant polytene labeling.
See response to F.
F) Show higher mag (e.g. insets) of polytene labeling, hard to see bands / interbands in current figures.
We have now added high magnification images of a portion of histone mutant polytene chromosome arms in Figure 1E.
2) Figure 1B: It would have been more complete to analyze S phase fraction by whole mount EdU labeling given that polytene squash immunolabeling can be highly variable.
We agree that EdU labeling is a more direct measure of DNA replication, but in our hands the cytology after EdU labeling is not necessarily less variable than with PCNA staining, which is easier to perform. Many studies have shown that PCNA staining is an excellent proxy for active replication in both Drosophila salivary glands and other cell types (e.g. see PMID: 30421640). Furthermore, we included HP1a staining in all polytene preparations to ensure that the chromosome preparations were not refractory to antibody staining. All PCNA positive polytene spreads were also positive for HP1a staining on telomeres, arms, and/or chromocenter (note that the H3K9R chromocenter does not stain well with HP1a).
3) Figure 2A: Make more explicitly clear that copy number in H3K9R is normalized relative to wild type salivary glands. The reader may interpret gray dots as rereplication of pericentric heterochromatic DNA above the copy number of euchromatic DNA, rather than less under-replication.
We added the following sentence to the text on page 7: “The data were plotted as a log2 transformed mutant/HWT ratio of normalized read counts, resulting in a relative copy number change for each 10kb window.”
4) Related to 3, can something be said about the absolute copy number of heterochromatic DNA in H3K9R? Is it close to fully duplicated copy number of euchromatic DNA? Was it normalized to control embryonic diploid reads? Copy number of some regions increased 16-fold relative to wild type, but it is unclear which loci those are and how much they are under-replicated in wild type.
Since we did not normalize our sequencing data to an spiked in external sample, we cannot determine the absolute copy number relative to a diploid cell. We indicated that some regions are elevated 16 fold just to give the reader a sense of the magnitude of the change based on our relative measurements, and all of these regions are within the pericentromere. We have also now included RPM values for chromosomes 2 and 3 in Figure S1 in which under-replicated domains are apparent.
5) Say whether all types of pericentric heterochromatic DNA was considered in the sequence analysis. If simple repeats were included, say how these highly repetitive DNA classes were bioinformatically analyzed.
We filter our sequencing read alignments using a MAPQ score of 10, which removes reads with low confidence mappability that often include simple repeats. Thus, the pericentric heterochromatin that we analyzed contain uniquely mapping reads. We have indicated this point in the Materials and Methods. In addition, our presentation of data as a ratio of mutant/control thus eliminates problems associated with poor read mappability. These strategies were used in our three previous publications (Penke et al. 2016, 2018, and Armstrong et al. 2018).
6) If HP1 requires H3K9me3 for localization, and HP1 and Su(UR) localization are co-dependent, why does Su(UR) affect IH underreplication whereas H3K9me3 does not? Is HP1 found in intercalary heterochromatin?
HP1a is retained on chromosome arms in H3K9R mutants (Penke et al. 2016); it is only lost from pericentric heterochromatin in H3K9R mutants.
7) Include error bars and p value asterisks within graphs of figure 1. Which classes were significantly different from each other in 1D?
As noted in response to reviewer 1, we considered the distribution of total counts among categories as a whole rather than considering individual categories separately. Thus we used a Chi squared test to ask if the distribution of the entire data set was different than expected, using the values in the HWT control as the expected values.
8) In several places “ectopic replication” is mentioned. I don’t think that’s the right term because what is being described is extra replication or changed timing, whereas ectopic means “abnormal place.”
We agree and have replaced “ectopic” with “more extensive” to refer to this replication.
9) Figure 1 legend says the chromocenter was identified by anti-HP1 labeling, but I am assuming that was not done for the mutants since H3K9R disrupts HP1 localization.
That is correct and thank you for pointing this out. The H3K9R mutant chromocenter was identified cytologically (which can be done for the HWT and H4K16R genotypes as well). As noted above, we routinely included HP1a staining in all polytene preparations to ensure that the chromosome preparations were not refractory to antibody staining. In H3K9R mutants we detect HP1a at telomeres and arms as we previously reported (Penke et al. 2016).

Reviewer 3 Report
In the article “H3K9 promotes under-replication of pericentromeric heterochromatin in Drosophila salivary gland polytene chromsomes”, Armstrong et al. establish the role of H3K9 in determining under-replication at the pericentromere. This point is demonstrated through the increased levels of ploidy along the pericentromere in H3K9 mutant drosophila. We recommend that this manuscript is accepted to Genes after the following revisions are made.
Major issues
1. PCNA experiment – it is unclear how cells were organized into S phase quintiles. Line 209 cites two articles which have used this method, both of which also don’t explain the method for categorizing pcna stained cells into different parts of S phase. This point is important since without being able to assess the accuracy of the method, it is hard to appreciate the changes between the mutants.
2. Figure 1C – as the authors stated in line 177 there is a developmental delay in the H3K9R mutant and thus the size differences shown in Figure 1C are probably due to this delay and have nothing to do with the improper cell cycle mentioned earlier.
3. Figure 2D – the decrease in the arms is most likely an artefact due to the strong increase in the pericentromeres. This frequently happens in sequencing based techniques, since an increase in one region results in less reads for the other regions since the total number of reads received by the sequencer is constant. This should be mentioned when discussed in lines 258, 296, and 308.
4. Figure 2E – these images are unclear and should be shown in a quantifiable way.
5. It would be helpful to add a paragraph to the discussion regarding the functional outcome of losing the pericentromeres under-replication.
Minor issues:
1. Line 109 – this sentence is not written clearly.
2. Line 173+174 – says figure 1C instead of 1B
3. Line 178 – says figure 1B instead of 1C
4. Line 191 – The legend states that “All S phase measurements were taken at the pre-wandering stage”. This is not relevant for figure 1C, which includes multiple stages, and is likely meant to be in 1A.
5. Figure 2C can be moved to the supplementary.
6. Line 246 – this should say “separated into” instead of “separated by”
7. The types of statistical tests used are not noted on lines 254, 294, and 296.
8. Line 305 – a “complete loss” is a strong conclusion. It is very likely that the under-replication may still exist at a lower level. This conclusion assumes that the CNVnator software is perfect although we know that it has false positive and false negative results as do all genomic tools. This finding should be worded more cautiously.
9. Line 333 – An increase specifically in earlier replication does not prove that the timing is condensed, as in this case all parts of replication would be sped up proportionally.
10. Line 367-8 – The suggestion that the k9R mutant effects under-replication by affecting the recruitment of suUR to pericentromeres is easy to check experimentally by IF. Adding it would upgrade the paper since it will give a mechanistic explanation to the results.
Author Response
1. PCNA experiment – it is unclear how cells were organized into S phase quintiles. Line 209 cites two articles which have used this method, both of which also don’t explain the method for categorizing pcna stained cells into different parts of S phase. This point is important since without being able to assess the accuracy of the method, it is hard to appreciate the changes between the mutants.
We agree with the reviewer’s request for an explicit description of the PCNA staging criteria. These criteria were added to the Methods section: “Salivary gland polytene chromosome immunofluorescence” and an in-text citation to the Materials and Methods section was included on line 225.
2. Figure 1C – as the authors stated in line 177 there is a developmental delay in the H3K9R mutant and thus the size differences shown in Figure 1C are probably due to this delay and have nothing to do with the improper cell cycle mentioned earlier.
We agree with the reviewer’s concern regarding the influence of the developmental delay of H3K9R mutants on our conclusions regarding the cell cycle phasing. We were cautious about using language that overinterpreted our data, using concluding phrases like “H3K9 is required for proper endocycle regulation” rather than making direct conclusions about S phase progression.
3. Figure 2D – the decrease in the arms is most likely an artefact due to the strong increase in the pericentromeres. This frequently happens in sequencing based techniques, since an increase in one region results in less reads for the other regions since the total number of reads received by the sequencer is constant. This should be mentioned when discussed in lines 258, 296, and 308.
We agree with the reviewer regarding this point and tried to be judicious in our interpretations in the original version by tempering our conclusions about changes observed along chromosome arms. We have now eliminated any biological conclusion and the following sentence was added to the text to be more explicit about this point: “This decrease may result from a larger proportion of H3K9R mutant reads mapping to pericentric regions, causing a corresponding relative decrease in the number of reads for other genomic regions, and thus is not likely biologically meaningful”(lines 275-277). Similar qualifying statements were included at lines 314-315 and 330 later in the manuscript.
4. Figure 2E – these images are unclear and should be shown in a quantifiable way.
A quantification of chromocenter organization in HWT and H4K16R genotypes has been added to Figure 2 (new Figure 2E). We cite our previous quantification of H3K9R in Penke et al. 2016. Further, all chromocenter images are now set to a similar scale rather than set to fill the image to emphasize the size increase in H3K9R mutant chromocenters relative to HWT or H4K16R.
5. It would be helpful to add a paragraph to the discussion regarding the functional outcome of losing the pericentromeres under-replication.
This is an interesting point that we chose not to speculate on, as there are no good biological assays for, or data on, the function of under-replication to our knowledge. In fact, Rif1 null mutants, which have no under-replication, are viable. Thus, rather than speculate we simply added a statement to this effect in the discussion (line 377-380).
Minor issues:
1. Line 109 – this sentence is not written clearly.
This sentence has been re-written: “We recently demonstrated that, in contrast to mutation of H3K9 writers, readers, and erasers, H3K9R mutant Drosophila diploid wing imaginal discs have only a modestly reduced S phase index, suggesting a small role played by H3K9 in canonical S phase progression”
2. Line 173+174 – says figure 1C instead of 1B
This change has been made.
3. Line 178 – says figure 1B instead of 1C
This change has been made.
4. Line 191 – The legend states that “All S phase measurements were taken at the pre-wandering stage”. This is not relevant for figure 1C, which includes multiple stages, and is likely meant to be in 1A.
This statement was moved to Figure 1B. Thank you for the correction.
5. Figure 2C can be moved to the supplementary.
A supplementary figure was created in response to a suggestion made by reviewer 3 and Figure 2C was moved to Supplementary Figure_S1.
6. Line 246 – this should say “separated into” instead of “separated by”
This change has been made.
7. The types of statistical tests used are not noted on lines 254, 294, and 296.
The Student’s T test has been included in the text at each of these lines.
8. Line 305 – a “complete loss” is a strong conclusion. It is very likely that the under-replication may still exist at a lower level. This conclusion assumes that the CNVnator software is perfect although we know that it has false positive and false negative results as do all genomic tools. This finding should be worded more cautiously.
We agree with the reviewer’s suggestion and have changed “complete loss” to “strong suppression.”
9. Line 333 – An increase specifically in earlier replication does not prove that the timing is condensed, as in this case all parts of replication would be sped up proportionally.
Agreed. We have changed the text to refer to the reduced S phase index. In addition, we consider this a “proposal” rather than a conclusion.
10. Line 367-8 – The suggestion that the k9R mutant effects under-replication by affecting the recruitment of suUR to pericentromeres is easy to check experimentally by IF. Adding it would upgrade the paper since it will give a mechanistic explanation to the results.
We completely agree that identifying the chromosomal localization of SuUR in an H3K9R mutant background would be a great experiment. Unfortunately, we have attempted this experiment many times using a SuUR antibody but failed to get this antibody to work on polytene chromosome spreads. We are in the process of generating a recombinant genotype with the H3K9R mutation and a GFP-tagged SuUR allele, but cannot complete this experiment in the time frame requested by the editor. As suggested by reviewer 1, we have added a citation indicating that SuUR localization to pericentric heterochromatin is reduced in HP1 mutants, consistent with SuUR recruitment to this region of the genome being dependent on H3K9.

Reviewer 4 Report
The authors extended their previous studies (Armstrong et al. 2018 Genome Research) to salivary gland polytene chromosomes and showed that histone mutation H3K9R but not H4K16R impaired under-replication of pericentromeric heterochromatin. Their results are clear and the conclusions are convincing. However, I would like to ask the authors to respond to the following concerns before publication.
Major points:
For the people outside of this field, it is better to indicate the region of pericentromeric heterochromatin in Figure 2A and B. If it is possible, it is nice to see the comparison between RPM and histone modification at H3K9 and H4K16 in Figure 2A and B.
The authors found that reduction of the DNA copy number at some regions in chromosome arms in H3K9R mutants as compared to the wild-type control: Figure 2A,D (lines 258-260), Figure 3B (lines 294-296), and Figure 3C,D (lines 306-310).These data suggest that H3K9 promotes replication at some arm regions. It is interesting to see whether those regions are enriched in H3K9 methylation or acetylation or some other modification. It is also interesting to know whether those regions are gene-rich or gene-poor. I would like to see how the authors discuss the mechanism by which H3K9R not just increased the DNA copy number in pericentromeric heterochromatin but also reduced the copy number in chromosome arms.
Minor points:
‘Figure 1C’ in line 173 must be ‘Figure 1B’.
‘Figure 1B’ in lines 177 and 178 must be ‘Figure 1C’.
Although the authors wrote in lines 252-253 ‘all major chromosome scaffolds (2L, 2R, 3L, 3R, 4, and X) between H3K9R or 253 H4K16R and HWT control (Figure 2A,B)’, the chromosomes 4 and X data are missing in Figure 2A,B.
The authors wrote ‘under-replicated (UR) domains’ in lines 279-280. But, it is better to show the abbreviation ‘UR’ when it appears for the first time: ‘under-replicated domains’ in line 277.
‘replication dependent H3’ in line 352 must be ‘replication-dependent H3’.
In the list of references, the first name of the authors is sometimes spelled out but sometimes not. It is better to follow the instruction of the journal.
Would you provide all the details of papers (volume, issue, page numbers) in references 26, 29, 31, 32, 53, and 56.

Author Response
Major points:
For the people outside of this field, it is better to indicate the region of pericentromeric heterochromatin in Figure 2A and B.
Cartoon diagrams of the metacentric Drosophila chromosomes 2 and 3 have been added to Figures 2A and B to depict the location of pericentric heterochromatin relative to genomic position.
If it is possible, it is nice to see the comparison between RPM and histone modification at H3K9 and H4K16 in Figure 2A and B.
H3K9me2 signal and RPM values for HWT, H3K9R, and H4K16R on Chromosomes 2L, 2R, 3L, and 3R have been added to Supplementary Figure_S1. Further, H3K9me2 signal has been added to Figures 2A and B as requested. We chose not to include H4K16ac signal as there is no strong enrichment to a particular chromosome region and found H3K9me2 signal to be more relevant to the observed under-replication phenotype.
The authors found that reduction of the DNA copy number at some regions in chromosome arms in H3K9R mutants as compared to the wild-type control: Figure 2A,D (lines 258-260), Figure 3B (lines 294-296), and Figure 3C,D (lines 306-310).These data suggest that H3K9 promotes replication at some arm regions. It is interesting to see whether those regions are enriched in H3K9 methylation or acetylation or some other modification. It is also interesting to know whether those regions are gene-rich or gene-poor. I would like to see how the authors discuss the mechanism by which H3K9R not just increased the DNA copy number in pericentromeric heterochromatin but also reduced the copy number in chromosome arms.
We agree that an enhancement of under-replication along chromosome arms in H3K9R mutants would be interesting. However, as Reviewer 3 pointed out, the decrease in copy number we report for H3K9R mutants most likely reflect a sequencing artefact rather than biology. We were worried about this issue in our initial submission and tried to temper our conclusions about changes observed along chromosome arms. Consequently, rather than delve further into copy number decreases bioinformatically, we followed the suggestion of Reviewer 3 to explicitly state the possibility of a sequencing artefact in the text (See response to Reviewer 3).
Minor points:
‘Figure 1C’ in line 173 must be ‘Figure 1B’.
This change has been made.
‘Figure 1B’ in lines 177 and 178 must be ‘Figure 1C’.
This change has been made.
Although the authors wrote in lines 252-253 ‘all major chromosome scaffolds (2L, 2R, 3L, 3R, 4, and X) between H3K9R or 253 H4K16R and HWT control (Figure 2A,B)’, the chromosomes 4 and X data are missing in Figure 2A,B.
We agree that the in-text citation in lines 252-253 was misleading and have changed the citation to Figure 2C. Figures 2A and B are representative chromosome arms for visualization purposes whereas the quantification in Figure 2C includes all major chromosome scaffolds as stated in the text.
The authors wrote ‘under-replicated (UR) domains’ in lines 279-280. But, it is better to show the abbreviation ‘UR’ when it appears for the first time: ‘under-replicated domains’ in line 277.
We have chosen not to use the “UR” abbreviation and have removed it from the text.
‘replication dependent H3’ in line 352 must be ‘replication-dependent H3’.
This change has been made.
In the list of references, the first name of the authors is sometimes spelled out but sometimes not. It is better to follow the instruction of the journal.
All author first (and middle) names are now abbreviated throughout the reference section.
Would you provide all the details of papers (volume, issue, page numbers) in references 26, 29, 31, 32, 53, and 56.
The volume, issue, and page numbers have been updated when available.
